# Uncertainty-Driven Semantic Segmentation through Human-Machine Collaborative Learning

**Mahdyar Ravanbakhsh**                    MAHDYAR.RAVAN@GINEVRA.DIBE.UNIGE.IT
*University of Genova, Genova, Italy*

**Tassilo Klein**                    TASSILO.KLEIN@SAP.COM
*Machine Learning Research, SAP SE, Berlin, Germany*

**Kayhan Batmanghelich**                    KAYHAN@PITT.EDU
*University of Pittsburgh, Pittsburgh, PA, USA*

**Moin Nabi**                    M.NABI@SAP.COM
*Machine Learning Research, SAP SE, Berlin, Germany*

## Abstract

Deep learning-based approaches achieve state-of-the-art performance in the majority of image segmentation benchmarks. However, training of such models requires a sizable amount of manual annotations. In order to reduce this effort, we propose a method based on conditional Generative Adversarial Network (cGAN), which addresses segmentation in a semi-supervised setup and in a human-in-the-loop fashion. More specifically, we use the discriminator to identify unreliable slices for which expert annotation is required and use the generator in the GAN to synthesize segmentations on unlabeled data for which the model is confident. The quantitative results on a conventional standard benchmark show that our method is comparable with the state-of-the-art fully supervised methods in slice-level evaluation requiring far less annotated data.

**Keywords:** Generative Adversarial Networks, Human-Machine Collaboration.

## 1. Introduction

Semantic image segmentation, which aims at assigning a class label to each pixel in an image, is one of the main applications of machine learning in medical image processing. Lately, deep learning techniques have been shown to attain exceptional results in this domain, outperforming the traditional approaches. However, large amounts of manually labeled data - which are key for supervised deep learning applications - are often expensive or impractical. To capitalize on the effectiveness of deep learning approaches for semantic segmentation tasks, while at the same time dealing with the limited availability of labeled data in the medical field, we propose a human machine collaboration (Abad et al., 2017) framework for medical image segmentation based on the popular generative adversarial network (GAN) framework. We show that the scores produced by the adversarial discriminator, which is trained to detect out-of-distribution samples, can be interpreted as inherent uncertainty estimates for active learning. The ability to directly use the adversarial discriminator score as a measure of uncertainty results in a convenient end-to-end approach to active learning. (Luc et al., 2016) propose the combination of cross-entropy and adversarial losses for semantic segmentation. (Souly et al., 2017) perform semi-supervised image segmentation

---

**Algorithm 1:** Collaborative Learning

---

**Input:** $I_{labeled}$, $I_{unlabeled}$, $k$, $n$
$G, D \leftarrow initialize()$, $S \leftarrow I_{labeled}$,
  $P \leftarrow I_{unlabeled}$
**for** $i \in 1..n$ **do**
  $\quad G, D \leftarrow train(S, G, D)$
  $\quad Q \leftarrow top(rank(D(P)), \frac{k}{n})$, $P \leftarrow P \setminus Q$
  $\quad S_{expert} \leftarrow humanExpert(Q)$,
  $\quad\quad S_{pseudo} \leftarrow G(P)$
  $\quad S \leftarrow S_{true} \cup S_{pseudo}$
**end**

---

Figure 1: Algorithmic (left) and concept view (right) of the proposed collaborative learning.

using GAN, leveraging unlabeled and generated data for estimating a proper prior. (Zhu and Bento, 2017) employ GAN for active learning for classification problems, generating samples to query rather than selecting them from a pool. None of these approaches use the discriminator score to measure model certainty.

## 2. Human-Machine Collaborative Learning with GAN

The proposed approach leverages conditional GAN (cGAN) for facilitating the human-machine collaboration for segmentation. To that end, the generator $G$ is trained to produce accurate label maps corresponding to the conditioned image, while the discriminator $D$ attempts to recognize whether a given segmentation is in accordance with the input image. What is more, $D$ can be used to estimate model uncertainty for unseen images. Specifically, we propose to use $D$ for ranking the predicted segmentations referred to as pseudo ground truth, such that annotations querying is restricted to low-confidence items. Thus expert annotations are obtained in an active learning fashion for out-of-distribution samples only, therefore incurring minimal cost. The process of learning the model decomposes in several stages. First, a supervised base model is initialized and trained using the small set of labeled samples $I_{labeled}$. Second, the model is trained in an interactive fashion for $n$ iterations. In each iteration, segmentation predictions are computed for the remaining unlabeled images $I_{unlabeled}$, which is followed by ranking. The top $\frac{k}{n}$ samples from the ranked pool are selected and queried for expert annotation, where $k$ is the total annotation budget, yielding labeled set $S_{expert}$. All other samples from $P$ are segmented using generator $G$, resulting in the labeled set $S_{pseudo}$. Last step in each active learning cycle is an update of the model $G, D$. The full training procedure is illustrated in algorithm 1.

## 3. Experimental Results

The proposed method is evaluated based on 3D cardiovascular MR images from the HVSMR 2016 challenge (Pace et al., 2015). The set consists of ten axial, cropped volumes from ten different patients with ground truth annotations. The images are segmented according to three labels: background, ventricular myocardium, and blood pool. The baseline method is a fully supervised cGAN employing a U-Net with skip connections as the generator network and a PatchGAN as the discriminator (Isola et al., 2017). The model is trained

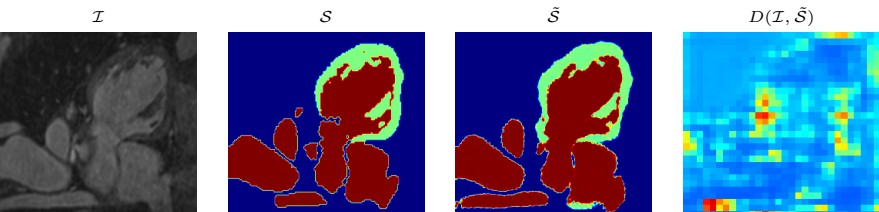

Figure 2: Original slice $\mathcal{I}$, ground truth $\mathcal{S}$, predicted segmentation $\tilde{\mathcal{S}}$, and discriminator score for predicted segmentation $D(\mathcal{I}, \tilde{\mathcal{S}})$. Red areas indicate where the pseudo ground truth is unreliable.

and evaluated using 10-fold cross validation. The proposed approach is based on the same architecture as the baseline network, but trained as described in Sec. 2. Using the slices of a single patient volume a base model is trained. In order to estimate a lower bound of accuracy attained by the proposed method - which uses a fraction of the labeled data used for the fully supervised model - an experiment consisting of a single active learning cycle ($n = 1$) is conducted. For simplicity, active learning cycles are simulated by different fractions of annotations. The supervised base model was used to determine the set of queries $Q$, before training a new model from the joint set $I_{labeled} \cup Q$. The experiment was conducted repeatedly for different values of budget $k$ expressed in terms of share of the total available labeled data (0% ... 100%).

Table 1: Dice scores for different amounts of supervised data and different benchmark models: Isola (Isola et al., 2017), Yu (Yu et al., 2017), and Shahzad (Shahzad et al., 2016).

|  | Proposed | | | | | | | | | SOTA | | |
|  | 10% | 20% | 30% | 40% | 50% | 60% | 70% | 80% | 90% | Isola | Yu | Shahzad |
|---|---|---|---|---|---|---|---|---|---|---|---|---|
| Myocardium | 0.41 | 0.45 | 0.53 | 0.57 | 0.62 | 0.67 | 0.71 | 0.75 | 0.73 | 0.73 | 0.82 | 0.75 |
| Blood Pool | 0.86 | 0.88 | 0.89 | 0.90 | 0.91 | 0.92 | 0.92 | 0.94 | 0.95 | 0.94 | 0.93 | 0.89 |
| Average | 0.64 | 0.66 | 0.71 | 0.74 | 0.77 | 0.80 | 0.82 | 0.85 | 0.84 | 0.84 | 0.88 | 0.82 |

## 4. Discussion and Conclusion

First experiments suggest a strong and significant correlation between Dice score and discriminator score (r = 0.98, p-value < 0.001). As a result, the discriminator appears to be a good indicator of the quality of the label maps produced by the generator (see Fig. 2), justifying its interpretation as a measure of uncertainty. As shown in table 1, the performance of the proposed active learning approach increases with larger portions of data annotated interactively, reaching nearly the performance of the fully supervised benchmark methods after training with only 80% of the labels. Note that for these experiments, only one active learning cycle was conducted. More active learning loops as well as incremental update of the model suggest to further improve the performance. This is because incrementally learning from new annotations is likely to change the model's ranking and selection of samples, exploring the pool of unlabeled samples more diversely.

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
