# OpenReview forum: "Uncertainty-Driven Semantic Segmentation through Human-Machine Collaborative Learning"
_MIDL.io/2019/Conference/Abstract — MIDL Abstract 2019_

### Official Review · AnonReviewer2 · 2019-04-29
**Sensible active-learning semi-supervised segmentation approach, experiments not extensive and hard to follow**

**Rating:** 3
**Confidence:** 3

**Review:**

The paper proposes to train a segmentation network and a discriminator. The scores of the discriminator on unlabeled images are used to select a subset of images for annotation by an expert. The segmentation output on the remaining unlabeled images are used as additional pseudo-labeled training images. This is a straightforward setup for semi-supervised active learning. It would be good to use these terms in title and abstract.

It is not clear to me if the discriminator is used to improve the U-net segmentation network, or if the networks are trained completely separately.

There is a lot of similar work that is not refered to. A useful additional reference would be https://arxiv.org/abs/1802.07934 but a search would find many more.

The algorithm in Figure 1 is not precise. S_true should probably be S_expert. I assume the manual segmentations are accumulated throughout all iterations, but the notation does not clearly reflect that.

A weakness is that the experiments use 2D slice based segmentation for a 3D data set. Slice based segmentation is not optimal for 3D segmentation tasks as has been shown in many studies. The paper would also be stronger if multiple data sets were used. It is hard to draw any conclusion from this small experiment on this single dataset.

I could not completely follow the experimental setup, it would be nice if a few more details were provided. Does 10% mean only a single scan (1 of 10) was used for training, referred to as the "base model"? What is the difference between the "baseline" and the "base model"? Which entry in the table is the "baseline network"? What does 20% mean? 10% "base model" plus 10% selected slices from all other (8?) scans? Is a validation data set used? I would expect a comparison between randomly selecting a percentage of slices for additional human annotation versus the selection made by the discriminator, but I do not think that that experiment has been performed? Also, it would be interesting to see the effect of using multiple rounds (n>1) but only n=1 was used? It would also be interesting to see the effect of using/not-using the S_pseudo images for additional training.

---

### Official Review · AnonReviewer1 · 2019-05-01
**.**

**Rating:** 3
**Confidence:** 3

**Review:**

the paper proposes an active-learning based method for semi-supervised learning, where the adversarial discriminator output can be interpreted as uncertainty to be used for seeking supervision.

---

### Decision · Program_Chairs · 2019-05-06
**Acceptance Decision**

Accept